# Microfabrication of a Chamber for High-Resolution, In Situ Imaging of the Whole Root for Plant–Microbe Interactions

**DOI:** 10.3390/ijms22157880

**Published:** 2021-07-23

**Authors:** Lauren K. Jabusch, Peter W. Kim, Dawn Chiniquy, Zhiying Zhao, Bing Wang, Benjamin Bowen, Ashley J. Kang, Yasuo Yoshikuni, Adam M. Deutschbauer, Anup K. Singh, Trent R. Northen

**Affiliations:** 1Environmental Genomics and Systems Biology, Lawrence Berkeley National Laboratory, Berkeley, CA 94720, USA; lkjabusch@lbl.gov (L.K.J.); dmchiniquy@lbl.gov (D.C.); ashleykang@berkeley.edu (A.J.K.); amdeutschbauer@lbl.gov (A.M.D.); 2CBRN Defense and Energy Technologies, Sandia National Laboratory, Livermore, CA 94550, USA; 3Joint Genome Institute, Lawrence Berkeley National Laboratory, Berkeley, CA 94720, USA; zyzhao@lbl.gov (Z.Z.); bwang16@lbl.gov (B.W.); bpbowen@lbl.gov (B.B.); yyoshikuni@lbl.gov (Y.Y.); 4Engineering Directorate, Lawrence Livermore National Laboratory, Livermore, CA 94550, USA; singh46@llnl.gov

**Keywords:** imaging, rhizosphere, microfabrication, microscopy, plant–microbe interactions, fluorescently-tagged bacteria, GFP-like proteins

## Abstract

Fabricated ecosystems (EcoFABs) offer an innovative approach to in situ examination of microbial establishment patterns around plant roots using nondestructive, high-resolution microscopy. Previously high-resolution imaging was challenging because the roots were not constrained to a fixed distance from the objective. Here, we describe a new ‘Imaging EcoFAB’ and the use of this device to image the entire root system of growing *Brachypodium distachyon* at high resolutions (20×, 40×) over a 3-week period. The device is capable of investigating root–microbe interactions of multimember communities. We examined nine strains of *Pseudomonas simiae* with different fluorescent constructs to *B*. *distachyon* and individual cells on root hairs were visible. Succession in the rhizosphere using two different strains of *P. simiae* was examined, where the second addition was shown to be able to establish in the root tissue. The device was suitable for imaging with different solid media at high magnification, allowing for the imaging of fungal establishment in the rhizosphere. Overall, the Imaging EcoFAB could improve our ability to investigate the spatiotemporal dynamics of the rhizosphere, including studies of fluorescently-tagged, multimember, synthetic communities.

## 1. Introduction

The rhizosphere is the area around the plant root in soil where microorganisms are densely populated and dynamic interactions between the plants and microorganisms occur. These complex interactions and the assemblage of microorganisms are established, shaped, and maintained by the plant root. Plants exude photosynthetically fixed carbons as nutrient sources, secondary metabolites, and signaling molecules to populate root ecological niches with beneficial microorganisms [1,2,3]. This “rhizosphere effect” has important implications for plant growth and protections against biotic and abiotic stressors, and for geochemical carbon cycling in soil environments [4,5]. With the expected rise of severe, climate-change-related weather conditions such as drought and the growing need to increase plant productivity with sustainable agricultural practices, it is vital to gain a mechanistic understanding of the rhizosphere effect and engineer the rhizosphere to improve plant growth and resilience.

To understand the mechanisms of the highly dynamic processes occurring in the rhizosphere, in situ interrogation of the system with high spatiotemporal resolution is necessary. However, due to the underground nature of the root system and the sheer complexity of the microbiome in the highly heterogeneous soil environment, studying root and microbial interactions has been challenging. For example, in a typical rhizosphere analysis experiment, the plant is uprooted from soil in the field or a pot at a defined time point, and the microbiome and other relevant chemicals are sampled in a destructive manner. This practice of uprooting the plant is limiting because the spatial information on microbial community members is lost and the same plant generally cannot be sampled over time. Further, there are many confounding variables, such as chemical reactions with minerals in soil that complicate the analysis and make the studies less reproducible and relevant across different locations and environments. To overcome these experimental hurdles, researchers have designed various types of specialized devices such as rhizotrons and microfluidics devices to improve specific aspects of sampling, analytics, and manipulation of the rhizosphere system, albeit often deviating substantially from the natural system [6].

Fluorescent microscopy is a promising tool for noninvasive in situ imaging of the microbial and root interactions with high spatiotemporal resolution. Developing in situ rhizosphere imaging methods is made especially more relevant as the microbial community colonization of the root and the persistence and succession patterns are likely dynamic and dependent on the developmental stage of the plant. Better knowledge in this can help guide synthetic microbial community inoculation protocols to optimize plant productivity. A notable development of in situ imaging devices is a microfluidic root chip by Massalha et al., in which real-time imaging of the microbial colonization of *Arabidopsis thaliana* seedlings was captured by high-resolution fluorescent confocal microscopy [7]. The co-inoculation by the fluorescent protein expressing *Bacillus subtilis* and *Escherichia coli* showed preferential localization, where *B. subtilis* established more densely at the root tip while *E. coli* dispersed homogeneously around the root [7]. While microfluidic devices have shown great promises in rhizosphere research, the microscale dimension of the root chamber limits the samples to be young seedlings (up to about a week old for Arabidopsis) of the model plants [7]. Rhizotrons can support the growing plants in a chamber for a much longer period of time, but are not designed for high-resolution microscopy [8,9].

Recently, we described the development of fabricated ecosystems (EcoFAB) that can support the various model plants (*Arabidopsis thaliana*, *Brachypodium distachyon*, and *Panicum virgatum*) for 3–4 weeks by having the bigger root chamber, which can be modified for conventional microscopy by using a glass slide as the lower plate [10]. We demonstrated that the EcoFAB can generate reproducible plant physiology and exudation phenotypes by conducting the multilab experiments with the model grass *B. distachyon* [11]. While the EcoFAB gives much improved imaging capability over rhizotrons and the imaging of the plant root phenotypes such as root hair length and branching pattern have become much more accessible, imaging the microbial interactions with the root remains challenging [10]. One reason is that the current design of the EcoFAB is not suitable for imaging with high-magnification and numerical aperture (NA) objectives to interrogate the microscale interactions at high resolution. Another reason is that the root can easily go in and out of the focal plane with the extra chamber height, a problem that would be exacerbated by using high-magnification objectives with a thinner focal plane.

To overcome these imaging constraints, we modified our EcoFAB to be an imaging-specific device (‘Imaging EcoFAB’) that combined a larger root chamber to support longer growth times and the ability to image root and microbial interactions (Figure 1) [12]. By implementing the pillar structures in the root chamber, the effective chamber height for the root is reduced so that the root remains closer to the glass surface (Figure 1A) while maintaining the flow of media throughout the chamber. The polydimethylsiloxane (PDMS) device body is bonded with a thin cover glass to reduce the working distance between the sample and microscope objective. Consistent hydration and supplementation of nutrients become more challenging with the more mature plants transpiring at a greater rate and the thinner chamber profile reducing the volume. We modified the commercially available growth chamber with a watering port so that the Imaging EcoFAB is housed in a sterile environment with easy sterile additions of water or media (Figure 1D). Using this system with high-resolution confocal microscopy, we imaged the colonization of the fluorescently labeled soil bacteria *Pseudomonas simiae* to the *B. distachyon* root up to 21 days after the germination, demonstrating the Imaging EcoFAB’s potential for in situ, high-resolution imaging of the rhizosphere for the duration of the life cycle of the model plants.

## 2. Results

### 2.1. Imaging EcoFAB Design and the Modified Magenta Box for Better Sterility Control

Imaging EcoFAB is designed to improve the imaging capability by reducing the effective height of the root chamber so that the plant root grows closer to the cover glass. By adding the pillar structures with dimensions of 1.5 mm × 1.5 mm × 2 mm (D × W × H) and pillars separated by 300 µm from each other throughout the top of the root chamber (Figure 1A,B and Appendix A), the effective chamber height became 1 mm and the *B. distachyon* root was forced to grow closer to the cover glass (Figure 1C). The pillar structures (vs. a shorter chamber) allowed effective exchange of nutrients and chemicals across the root chamber and yielded plants with similar growth patterns to the original EcoFAB. The dimensions of the pillars and the distance between them can be modified based on the plant type (i.e., root thickness) and experimental specifications. However, the 3D printers have limitations with the aspect ratio of the 3D structures and resolution of the features. The printability of the pillars needs to be experimentally tested for each type of printer and resin (Appendix A). Once the mold is printed and treated for PDMS casting (see Methods and Appendix A), the PDMS body is bonded with the thin cover glass to reduce the distance between the objective and the root to accommodate high magnification objectives with the short working distance (Figure 1A).

To keep the Imaging EcoFAB sterile throughout the experiment, it was housed in a modified Magenta box (see Methods) where the media and water can be injected into the EcoFAB root chamber from the outside of the box through a 0.2 µm filter with luer lock fitting (Figure 1D). This minimized the Imaging EcoFAB’s exposure to outside environments and resulted in better control of sterility compared to the original Magenta box, which required opening of the lid to add media into the EcoFAB in a biosafety cabinet. During the imaging session, the imaging EcoFAB was taken out of the box to be placed on the microscope stage, sterility of the device was monitored by collecting the media in the chamber then plating onto the LB plate, which showed that the sterility was maintained during 14 days of examination. *B. distachyon* was routinely grown in the Imaging EcoFAB for 3 to 4 weeks (Figure 1C).

### 2.2. Imaging of the Entire B. distachyon Root at High Magnification

The sterile *B. distachyon* root in an Imaging EcoFAB was imaged on days 0, 2, 4, 7, 10, 14, 17, and 21 postgermination (Figure 2 and Appendix A). The image of the entire root was taken at different magnifications (10×, 20×, and 40×) by stitching the tile-patterned images using the microscope system’s automated stage and the software’s tile imaging setting (Figure 2 and Appendix A, see Methods). This revealed that mapping of the detailed structure of the entire live root at 3 weeks old or older is possible. However, due to the long duration it takes to cover the large area at high magnification—collecting the image of entire 21-day old root at 40× covering a 1300-mm^2^ area took 10 h (Figure 2)—this type of imaging is not suitable for the situation where high temporal resolution is necessary, such as studying the root colonization pattern of highly motile bacteria or fast-growing plants. Further, as the Imaging EcoFAB makes the entire root system flattened to the cover glass, effectively making it into a 2D structure, the entire root system is accessible to high-magnification and -resolution imaging (Figure 1A). Furthermore, since no dissection or root preparation is needed with the Imaging EcoFAB, temporal studies with the same plant are possible. The spatiotemporal colonization pattern of various bacterial species on the entire root can be effectively studied using this system.

### 2.3. Multispectral Imaging of Fluorescent Protein Expressing P. simiae on B. distachyon Root 

The ability to spectrally distinguish colocalizing bacteria in the rhizosphere is beneficial to interrogate the microbial interactions, especially while imaging at high magnification and resolution to allow segmentation and cell counting for quantitative data analysis. To demonstrate this in an Imaging EcoFAB, *P. simiae* strains engineered to constitutively express the fluorescent proteins mTagBFP, mTurquoise2, EGFP, mVenus, mKO, mApple, mCherry, mKate2, and mCardinal (a total of 9 strains expressing a different fluorescent protein) were inoculated into a week-old sterile *B. distachyon* [13]. After 3 days, the spectral images at 10×, 20×, and 40× magnification were acquired in the lambda mode of ZEN software as well as the linear unmixing of the spectral images (Figure 3). Due to the excitation and emission spectral overlaps, all 9 fluorescent proteins could not be distinctly visualized. We also observed that the blue emitting proteins mTagBFP and mTurquoise2 did not have detectable fluorescence signals, possibly due to slower growth due to greater metabolic burdens to express these proteins. We speculate that this may be due to the greater metabolic burdens to express these proteins. 

The spectral images analyzed using K-means clustering algorithm (scikit-image) and the fluorescence signals were clustered into four spectral categories (Appendix A): green (EGFP, mVenus), orange (mKO), red (mCherry, mApple, mCardinal, mKates2), and background (mTagBFP, mTurquoise2, background noise). These three colors are represented in the spectrally unmixed images from the lambda mode (Figure 3). In the 40× image, which is focused around root hairs (shown through autofluorescence represented by green color), the individual bacterial strains are resolved at various densities (Figure 3C). Using this image, the segmentation analysis was conducted by first generating the binary image with the fluorescence intensity threshold and then applying the dilation and erosion operations with the circular structuring element to separate out the individual bacterium (Appendix A). This initial analysis yielded the counting of 478 cells in the image. However, to improve the accuracy of cell counting in the rhizosphere, the 3D multispectral image using z-stack is necessary as the different orientations of the rod-shaped *P. simiae* yield either rod or circular shapes on 2D profile. With the improved software package of the microscope system that supports multispectral 3D segmentation using machine learning algorithm (i.e., Zen Intellesis package for the Zeiss confocal systems), the quantitative analysis of the rhizosphere microbiome will become more accurate and mainstream.

### 2.4. Bacterial Succession Study with Two Fluorescent P. simiae Strains

Imaging EcoFAB can be used for studying the bacterial persistence and succession patterns in the rhizosphere with the same live plant in situ over a 3- to 4-week duration. To examine this application, we conducted a succession study using two fluorescent strains of *P. simiae*. These two strains have identical genomes except for the fluorescent protein, allowing for the demonstration of a succession experiment in the device where a first one strain is established in the rhizosphere, followed by a second stain co-locating. First, mTagBFP expressing *P. simiae* was inoculated to the 2-week-old sterile *B. distachyon* at OD600 of 0.5. The higher concentration of mTagBFP strain (vs. 0.01 in the multispectral imaging, see Methods) was inoculated to compensate for its slow growth. After three days, establishment of the mTagBFP strain was confirmed by fluorescent imaging (Figure 4A). Then, the *P. simiae* with mCherry expression was inoculated to the same plant at OD600 of 0.1 and imaged after 2 days, showing the colocalization of mTagBFP and mCherry fluorescence around the root tip of *B. distachyon* (Figure 4B).

### 2.5. Imaging with Various Solid Substrates in the Root Chamber and Fungal Hyphae

Although a hydroponic system is advantageous for optical imaging of the rhizosphere, it deviates from the natural soil system’s solid structure for the root and microbiome. Optical imaging of rhizospheres in soil, however, is difficult due to its opaqueness and autofluorescence. Some microbiomes, notably many fungi, need solid structures for their growth and establishment around the root tissue. To test the Imaging EcoFAB’s imaging ability with the optically transparent solid substrates, the Imaging EcoFAB was filled with different solid substrates—sand and phytagel—for imaging *B. distachyon* root. Sterile quartz sand was filled in an Imaging EcoFAB, wetted with 10% MS media, and B. distachyon was grown for a week. A bright field imaging of the root at 40× shows the detailed structure of root hair along with the sand particles around the root (Figure 5A). Additionally, an Imaging EcoFAB was filled with 2 g/L phytagel (with 10% MS) and the *B. distachyon* was inoculated with the H1-RFP-expressing fungi, *Neurospora carassa* [14]. *N. carassa* did not grow well in the hydroponic system but showed growth in phytagel substrate. The image shows *N. carassa* hyphae interacting with *B. distachyon* root hair (Figure 5B). Agar (1%) with 10% MS in the Imaging EcoFAB had similar results to phytagel (results not shown).

## 3. Discussion

In this study, we improved the rhizosphere imaging ability of the original EcoFAB by adding pillar structures on the top of the root chamber and by using a thin cover glass as the bottom substrate. The device’s improvement was demonstrated by high-magnification and -resolution confocal imaging of the entire *B. distachyon* root and microbial interactions while maintaining the key advantages of the EcoFAB: real-time in situ imaging using conventional microscope system, bigger chamber to support the model plants for 3 to 4 weeks, ability to pack solid substrates, and improved environmental control such as sterility of the chamber. The rhizosphere is an area of high bacteria–plant interaction and a target for improving plant growth for agricultural and energy crops in marginal soil. However, the root system in soil is inherently difficult to study due to its complexity and many location-specific variables. To gain mechanistic insights into rhizosphere, laboratory-based devices with controlled environments and controllable microbiome communities are beneficial as they allow us to delve deeper into the interactions.

Massalha et al. co-inoculated the fluorescently tagged *E. coli* and *B. subtilis* in an *A. thaliana* root and observed the *B. subtilis* establishing close to the root tissue while *E. coli* is excluded away, hovering over the root, suggesting possible antagonistic interactions between *B. subtilis* and *E. coli* in rhizosphere [7]. To increase the relevance of such a reduced system to the natural system, at least the taxonomically and functionally representative members of the bacteria and fungi in the rhizosphere need be co-inoculated and their live interactions quantitatively measured. In general, the taxonomic diversity decreases from the bulk soil to rhizosphere, albeit with increased metabolic activities due to the selective force by the plants and antagonistic behavior of the root colonizers, with the most abundant bacterial taxa in rhizosphere to be *Proteobacteria*, *Acidobacteria*, *Actinobacteria*, *Furmicutes,* and *Bacteroidetes* depending on the plant types [15]. In this study, we demonstrated multispectral imaging of the *P. simiae* strains expressing 9 different fluorescent proteins. Due to the spectral overlaps among these fluorescent proteins however, not all 9 strains could be unmixed but were binned into 4 different spectral categories. To increase the number of spectrally distinct bacteria in rhizosphere, there needs to be various approaches such as protein engineering to narrow the excitation and emission bandwidth of the fluorescent proteins, utilizing fluorescent dyes or quantum dots, lifetime measurement to further delineate the fluorescent species, and development of a new line of fluorescent protein expression system to expand the spectral range such as phytochrome-based fluorescent proteins for near-infrared fluorescence [16,17]. With the continual development of the biosensors as well as the fluorescent protein expression system such as CRAGE-Duet that engineered the *P. simiae* strains in this study, the Imaging EcoFAB is poised to gain valuable spatiotemporal information of the relevant synthetic community in rhizosphere [13].

Another interesting area to explore using Imaging EcoFAB due to its larger chamber size is defining the boundary of the rhizosphere, whose definition has varied depending on the study. In experiments where plants are grown in natural soils, it is sometimes pragmatically defined as the soil attached to the roots; in general, many studies extend the rhizosphere for up to 4 mm [18,19,20,21,22]. However, attempts to define the rhizospheric boundaries have primarily taken a more holistic view, focusing on considerations of specific soil contexts (e.g., moisture) and the activity or properties of the chemical(s) of interest [18,23,24,25,26]. With improved methods of capturing the spatiotemporal dynamics of the rhizosphere in devices like the imagining EcoFAB, a third or perhaps unifying definition of the rhizosphere could arise. Paired with the use of simplified or synthetic communities of fluorescently tagged microorganisms in plant root imaging systems, the Imaging EcoFAB could help define the reaches and bounds of the rhizosphere.

We believe that a deeper understanding of the rhizosphere must include interrogating the spatiotemporal dynamics at high-resolution and quantitative data analysis. The future of these developments in EcoFAB with high-resolution imaging could easily be tapped into by expanding the root imaging phenotyping tools like MyROOT 2.0 and RootNav 2.0, and the imaging analysis packages like scikit-image [27,28,29]. Additionally, these sources of data could be fed as positional information into platforms such as COMETS for molecular understanding of rhizosphere dynamics [30]. The eventual goal is to gain the mechanistic understanding of the plant–microbial interactions in the rhizosphere and drive the discoveries in improved plant growth promotion.

## 4. Materials and Methods

### 4.1. Fabrication of the Imaging EcoFABs

The Imaging EcoFAB was designed using Solidworks 2018 SP2 (Solidworks, Waltham, MA, USA). The design of the device includes the pillar structures with dimensions of 1.5 mm × 1.5 mm × 2 mm (D × W × H) separated by 300-µm spacing between the pillars on the top of the root chamber to push the root close to the cover glass (Figure 1A). The negative mold was printed using a Form 2 3D printer from Formlabs with the proprietary clear resin version 4 (Formlabs, Somerville, MA, USA). The mold was washed in isopropanol for 20 min in a Formlabs Wash machine and further rinsed with fresh isopropanol using a wash bottle to remove uncured resin. After being air dried, the mold was further cured using a Formlabs Cure instrument for 30 min at 60 °C. Then, the extra support structures from the printing process were removed, and the rough remains on the mold were sanded off and further rinsed with deionized water. The mold was then dried at 80 °C for several hours before the casting process. The device file can be downloaded from https://eco-fab.org/device-design/.

Fabrication process of the Imaging EcoFAB is similar to the process described in Gao et al. for constructing the previous version of the EcoFABs [10]. Briefly, the 2 × 2 EcoFAB mold was lined with plastic packaging tape, then filled with 2.5 g of a 10:1 mixture of polydimethylsiloxane (PDMS) base to fixer (Dow Sylgard 184 Silicone Encapsulation Clear, Dow Chemical, Midland, MI, USA). After degassing the PDMS in the mold using a vacuum chamber for 30 min, the PDMS was incubated at 80 °C until it solidified, which took about 3 h. The PDMS was carefully peeled from the mold and trimmed into individual Imaging EcoFAB tops. The EcoFAB tops were covalently bonded to an ethanol-cleaned large cover glass (50 × 75 mm, thickness 0.13–0.17 mm, part #260462, Ted Pella, Redding, CA, USA) using a plasma cleaner with supplemental oxygen (Plasma Cleaner PDC-001, Harrick Plasma, Ithaca, NY, USA). The imaging EcoFABs were filled with sterile DI water immediately after bonding to keep the inside PDMS surface hydrophilic. EcoFABs were placed on a hot plate at 70 °C for 5 min to improve the bonding. Imaging EcoFABs were emptied of water, then placed in a modified magenta box for autoclave sterilization.

### 4.2. Magenta Box Housing Modification

The modified magenta box was constructed using a magenta box with a vented lid (MK5 with Vented Lid, Caisson labs, Smithfield, UT, USA). A 5/16” diameter bit was used to drill a hole in the upper right corner of the magenta box when placed on the largest side, 1 cm from the top and side. Inside, a luer lock fitting (Nylon, Straight, Male Luer Lock Ring × Female Luer, Cole Parmer, Chicago, IL, USA) was pushed through the hole, male end out. A rubber gasket (water- and steam-resistant high-temperature EPDM O-ring, 0.103” width, 0.237” ID, 0.443 OD, McMaster-Carr, Elmhurst, IL, USA) was placed on the exterior of the magenta box on the luer lock fitting. A second identical luer lock fitting was attached to the first luer lock fitting and tightened. A blunt tip needle (stainless steel dispensing needle with luer lock connection ½” needle length, 23 gauge, McMaster-Carr, Elmhurst, IL) was inserted into a 3.5 in (9 cm) piece of tubing (platinum-cured silicone, 1/50” ID × 1/12” OD, Cole Parmer, Chicago, IL, USA). On the other end of the tubing, a 0.75 in (1.9 cm) piece of stainless steel tubing (304 stainless steel tubing 0.025” OD, 0.006” wall thickness, McMaster-Carr, Elmhurst, IL, USA) was inserted. The luer-end of the silicone tubing assembly was tightened into the luer lock fittings on the magenta box.

### 4.3. Brachypodium Distachyon Seed Sterilization and Germination

The *Brachypodium distachyon* BL-21 seeds were kindly provided by Dr. John Vogel (Joint Genome Institute, Berkeley, CA, USA) [31]. The seeds were sterilized by submerging seeds with the awn peeled away in 70% ethanol/water for 30 s [11]. Then, the ethanol was drained and the seeds were submerged in 50% household bleach solution for 5 min. The bleach was drained and the seeds were rinsed 5 times with sterile milli-Q water. The seeds were arranged on a sterile 1% agar water plate. Seeds were germinated for 3 days in a growth chamber (25 °C, 16/8 light/dark cycle, 32% humidity, 604 PAR in the center of the unit, CMP6010, Coviron, Winnipeg, MB, Canada). Of the germinated seedlings, those with similar size and shape were selected and inserted into the sterilized and 10% Murashige and Skoog (MS)-media-filled EcoFAB via plant opening (Figure 1A).

### 4.4. Pseudomonas Simiae Inoculation

The plants were inoculated with *Pseudomonas simiae* WCS417r, engineered to express the fluorescent proteins [13]. Each line of *P. simiae* constitutively expresses mTagBFP, mTurquoise2, EGFP, mVenus, mKO, mApple, mCherry, mKate2, and mCardinal, totaling 9 fluorescent proteins. For inoculation of each *P. simiae* line, the overnight cultures were prepared in LB, pelleted, and resuspended in 10% MS media (3 replicates per line and no treatment). The final OD600 was measured and normalized using 10% MS media before the inoculation. For multispectral imaging, all 9 lines of *P. simiae* were inoculated to the 1-week-old sterile plants in the Imaging EcoFABs. Each fluorescent line was normalized to OD600 0.01 in the EcoFAB chamber after inoculation.

For *P. simiae* root establishment and succession pattern analysis, first, *P. simiae*–mTagBFP was inoculated on a 2-week-old sterile plant at OD600 0.5 in the chamber (3 replicates per treatment, including null). The inoculation cell concentration was high so as to simulate ecological niche saturation. After three days, the image of the root was taken. Then, *P. simiae*–mCherry was inoculated at a lower OD600 of 0.1, to imitate a new succession. The image was taken after two days of the second inoculation.

### 4.5. Neurospora Crassa Inoculation

Imaging EcoFABs were filled with quartz sand (Sigma-Aldrich, 50–70 mesh particle size) prior to autoclaving. Prior to planting, sand Imaging EcoFABs were filled to capacity with 10% MS. For gel-filled Imaging EcoFABs, 1.5 g/L phytagel was prepared in 10% MS, autoclaved, and filled into Imaging EcoFABs to capacity while still hot. As described above, *B. distachyon* seedlings were prepared and allowed to grow for 1 week prior to inoculation with fungal strains. The *Neurospora crassa* strain was kindly provided by Dr. N. Louise Glass (UC Berkeley, Berkeley, CA, USA) [14]. Plants were inoculated at 10^5^
*N. crassa* spores per mL of Imaging EcoFAB volume (three plants per treatment or media). Plants were grown in the growth chamber (setting described above) for 2 days prior to imaging.

### 4.6. Imaging and Analysis

The plant roots in the Imaging EcoFABs were imaged using a Zeiss LSM 710 confocal microscope system using Zen 2.3 SP1 software (Zeiss, Oberkochen, Germany). Before imaging, the EcoFAB device was removed from the magenta box and placed on a microscope stage in inverted configuration. Images were colorized through Zen 2.3 SP1 software for multichannel, overlaid images. Images were analyzed with scikit-image (version 0.18) [29]. From the Zeiss-specific propriety LSM file for the 40× multispectral image (Figure 3C), Kmeans was used to detect the number of distinct colors of proteins by clustering pixel-by-pixel to find bins of channels. The results were expressed in a binned dendrogram (Appendix A). The segmentation analysis was conducted also by using the 40× multispectral image (Figure 3C), with the dilation and erosion operations using the circular structural element applied to the image to separate out the individual bacterium. This analysis counted 478 cells in the image. The environment, code, and parent data file are available in the Appendix A.

## 5. Conclusions

The field of plant–microbe interactions requires advancements in fluorescent probes and imaging platforms to understand and dissect complex spatiotemporal dynamics in the rhizosphere. In this study, we developed the imaging EcoFAB, a device that supports a model plant *B. distachyon* for 3+ weeks and is suitable for whole-root, high-resolution and -magnification imaging using a conventional microscope system with a high degree of environmental control. Using this device, we imaged the fluorescently tagged bacterial and fungal interactions in the rhizosphere. The device accommodated solid substrates such as sand and agar in the root chamber, making it possible to study the microbes that do not grow well in a hydroponic system. Quantitative analysis of the multispectral image of the fluorescently tagged *P. simiae* strains demonstrated imaging EcoFAB’s potential to drive more reproducible and quantitative research of the field-relevant synthetic rhizosphere community. With the continual improvement of imaging systems, imaging analysis algorithms, and fluorescent probes, imaging EcoFAB can be a powerful platform to interrogate and dissect the complex plant–microbial community dynamics in the rhizosphere. 

## 6. Patents

The device currently has a pending patent in the United States, United States Patent Application 20200385663

## Figures and Tables

**Figure 1 ijms-22-07880-f001:**
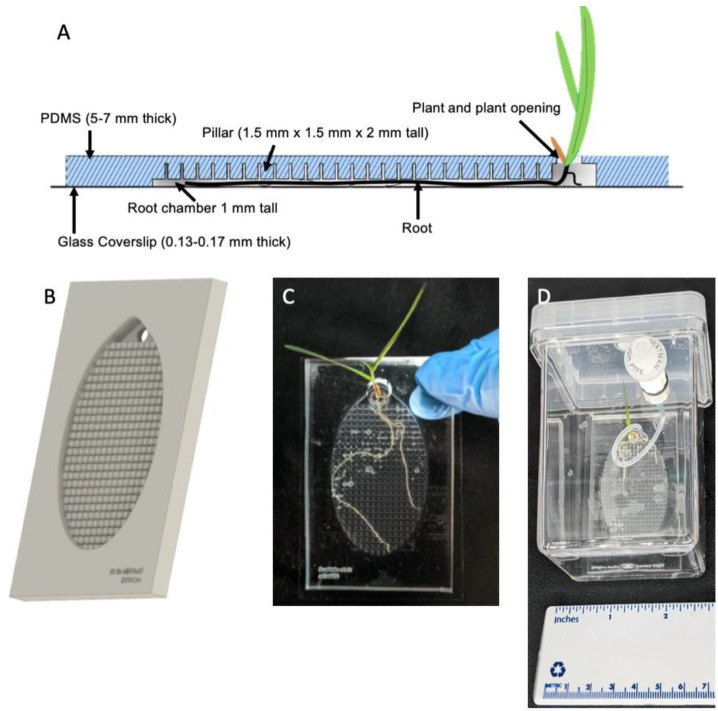
(**A**) Side-view of the Imaging EcoFAB device. The addition of pillars in the device allows media to flow easily through the device, while roots grow flat and against the coverslip. (**B**) Rendering of the PDMS component of the device. Shoots grow out of the top port. (**C**) Imaging EcoFAB with *B. distachyon* plant (20 days post germination, hydroponically grown in 0.1 MS). (**D**) Imaging EcoFAB in sterile housing with *B. distachyon* plant (14 days postgermination, hydroponically grown in 0.1 MS).

**Figure 2 ijms-22-07880-f002:**
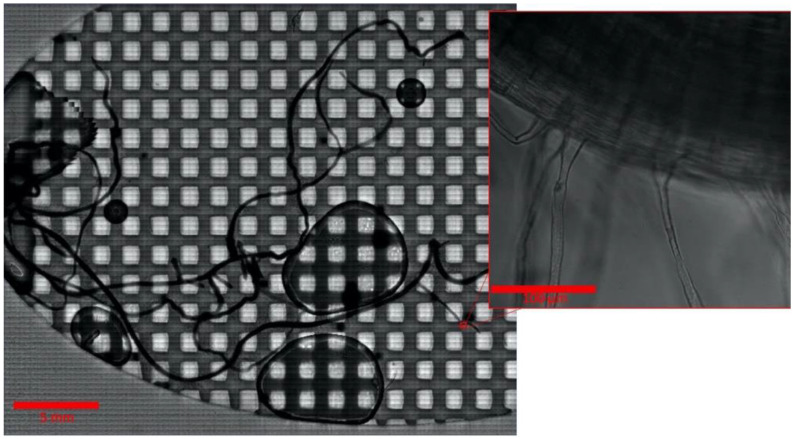
*B. distachyon* root after 21 days in an Imaging EcoFAB. The whole root is visible at high magnification with a composite scan. Inset is one field of the composite scan.

**Figure 3 ijms-22-07880-f003:**
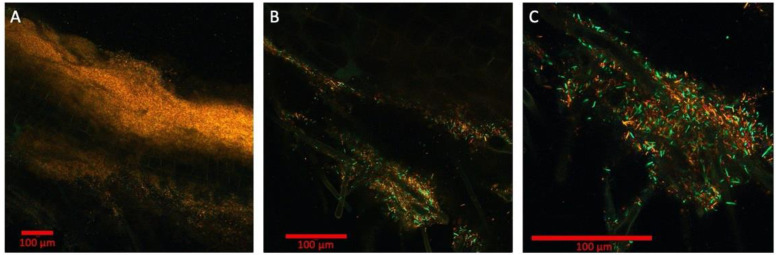
Multispectral imaging with 9 engineered strains of *Pseudomonas simiae* with constitutive expression of 9 fluorescent proteins: mTagBFP, mTurquoise2, EGFP, mVenus, mKO, mApple, mCherry, mKate2, and mCardinal. The magnifications shown are 10× (**A**), 20× (**B**), and 40× (**C**), respectively. Bars are 100 μm. Colorized in Zen (Zeiss).

**Figure 4 ijms-22-07880-f004:**
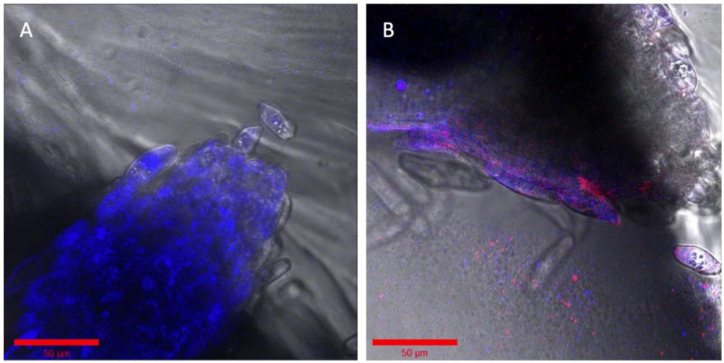
Succession of two similar engineered strains of *Pseudomonas simiae*. (**A**) The root was incubated with OD600 0.5 of *P. simiae* with mTagBFP in the imaging EcoFAB chamber for 3 days. (**B**) The same root is shown inoculated with mCherry—*P. simiae* after incubation for two days. Bar is 50 μm in both images. Colorized in Zen (Zeiss).

**Figure 5 ijms-22-07880-f005:**
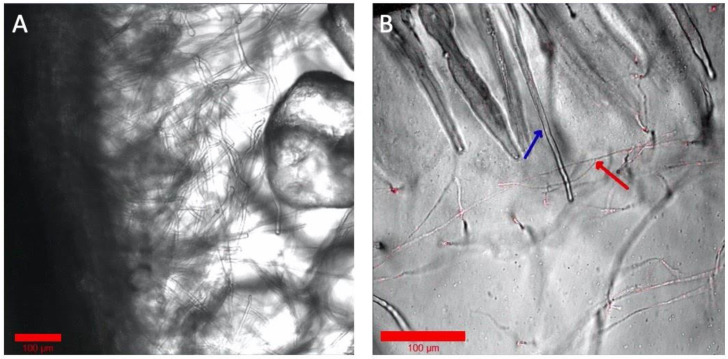
Examination of solid and gel media types in Imaging EcoFABs. (**A**) Axenic *B. distachyon* grown on sand, watered with 10% MS. (**B**) *B. distachyon* grown wih H1-RFP *N. crassa* in phytagel with 10% MS. The blue arrow shows a root hair and the red arrow shows hyphae. Colorized and annotated in Zen (Zeiss).

## Data Availability

The 3D-printing-ready design files of the EcoFAB are freely available at http://eco-fab.org/device-design/. The data presented in this study are available in this article and Appendix A.

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
