# Peer review of "Microfabrication of a Chamber for High-Resolution, In Situ Imaging of the Whole Root for Plant–Microbe Interactions"

_ijms, 2021, doi:10.3390/ijms22157880_

Round 1
Reviewer 1 Report
Fabricated ecosystems (EcoFABs) offer an innovative approach to in situ examine microbial establishment patterns of plant roots using non-destructive, high-resolution microscopy. In this article paper, Dr. Peter W. Kim, Dr. Trent R. Northen, and co-workers describe a new ‘Imaging EcoFAB’ device to image the entire root system of growing Brachypodium distachyon at high resolution. The device applies to investigating root-microbe interactions of multi-member communities and improve our ability to investigate the Spatio-temporal dynamics of the rhizosphere.
The manuscript's general style and overall representation are well organized, and the research pattern and details are described with accuracy. Furthermore, the authors need to address the following concerns and make a major revision before this manuscript is accepted by Int. J. Mol. Sci for publication
Particular comments:
- The part of the summary/conclusion is absent.
- Multiple places were identified that reference citation forms are not based on the author guidelines of the specific journal. For example, DOI mistakes for Ref. 10, 22; page numbers for Ref. 7, 14, 20; Ref. 12, the patent format is incorrect. The senior author needs to check all the cited references one by one before proofreading the revision.
Author Response
Response to Reviewer 1 Comments
Thank you for taking the time to review the manuscript and the feedback. Here are the brief responses to the feedback.
Point 1: The part of the summary/conclusion is absent.
Response 1: We added a separate conclusion section 5 in the edited draft.
Page 11, line 406-421: “The field of plant-microbe interactions requires advancements in fluorescent probes and imaging platforms to understand and dissect complex spatiotemporal dynamics in the rhizosphere. In this study, we developed the imaging EcoFAB, a device that supports a model plant B. distachyon for 3+ weeks and is suitable for whole-root, high resolution and magnification imaging using conventional microscope system with the high degree of environmental control. Using this device, we imaged the fluorescently tagged bacterial and fungal interactions in the rhizosphere. The device accommodated solid substrates such as sand and agar in the root chamber, making it possible to study the microbes that don’t grow well in hydroponic system. Quantitative analysis of the multi-spectral image of the fluorescently tagged P. simiae strains demonstrated imaging EcoFAB’s potential to drive more reproducible and quantitative research of the field-relevant synthetic rhizosphere community. With the continual improvement in imaging system, imaging analysis algorithm, and fluorescent probes, imaging EcoFAB can be a powerful platform to interrogate and dissect the complex plant-microbial community dynamics in rhizosphere.”
Point 2: Multiple places were identified that reference citation forms are not based on the author guidelines of the specific journal. For example, DOI mistakes for Ref. 10, 22; page numbers for Ref. 7, 14, 20; Ref. 12, the patent format is incorrect. The senior author needs to check all the cited references one by one before proofreading the revision.
Response 2: Thank you for pointing this out. We made the necessary correction for the reference formats, and corrected and added DOI info to all the references in the revised draft.
Reviewer 2 Report
The manuscript includes very interesting original data that can be of great practical importance.
There are several issues that need to be corrected and explained:
- Section 4. Materials and Methods, subsection 4.6. Imaging and Analysis
The number of repetitions should be indicated here.
- Do the authors have detailed results of quantitative data analysis? What are the accuracies for segmentation and cell counting?
- A separate section Conclusions should be added.
Author Response
Response to Reviewer 2 Comments
Thank you for taking the time to review the manuscript and the valuable feedback. We have revised the manuscript to address your comments as summarized below.
Point 1: - Section 4. Materials and Methods, subsection 4.6. Imaging and Analysis
The number of repetitions should be indicated here.
Response 1: Thank you for pointing this out. This has now been corrected as described below. Most of the images had three replicates for each condition. Specifically for section 4.6, the quantitative imaging analysis was conducted using the 40x image from Figure 3C.
In section 4.6, it now indicates that the kmeans clustering analysis and segmentation analysis were conducted using the Figure 3C.
Page 10, line 398 to 400: “From the Zeiss specific propriety LSM file for the 40x multi-spectral image (Figure 3C), Kmeans was used to detect the number of distinct colors of proteins by clustering pixel-by-pixel to find bins of channels.”
Page 10, line 401-402: “The segmentation analysis was conducted also by using the 40x multi-spectral image (Figure 3C),”
The number of the replicates are indicated in the Materials and Methods section now as shown below
Page 10, line 368: “(3 replicates per line and no treatment)”
Page 10, line 374: “(3 replicates per treatment, including null)”
Page 10, line 388: “(three plants per treatment or media)”
Point 2: Do the authors have detailed results of quantitative data analysis? What are the accuracies for segmentation and cell counting?
Response 2: Thank you for pointing this out. While performing quantitative analysis was not the focus of the manuscript, we have included preliminary results in the SI and noted the limitations of this analysis in the revised manuscript.
The result section 2.3 now addresses the segmentation and cell counting analysis.
Page 5, line 184 to 194: “Using this image, the segmentation analysis was conducted by first generating the binary image with the fluorescence intensity threshold, and applying the dilation and erosion operations with the circular structuring element to separate out the individual bacterium (Figure S4). This initial analysis yielded the counting of 478 cells in the image. To improve the accuracy of the cell counting in rhizosphere, however, the 3D multi-spectral image using z-stack is necessary as the different orientations of the rod-shaped P. simiae yield either rod or circular shape on 2D profile. With the improved software package of the microscope system that supports multi-spectral 3D segmentation using machine learning algorithm (i.e. Zen Intellesis package for the Zeiss confocal systems), the quantitative analysis of the rhizosphere microbiome will become more accurate and mainstream.”
We included the Figure S4, showing the result of segmentation analysis using the circular structural element as the basis.
In Appendix A:
Figure S4. The segmentation analysis of the multi-spectral image of P. simiae (Figure 3C) from scikit-image package. The fluorescence signals were dilated and eroded using the circular structuring element to help distinguish the individual bacterium. The segmentation analysis counted the total of 478 cells in the image.
Point 3: A separate section Conclusions should be added.
Response 3: We added a separate conclusion section 5 in the edited draft.
Page 11, line 406-421: “The field of plant-microbe interactions requires advancements in fluorescent probes and imaging platforms to understand and dissect complex spatiotemporal dynamics in the rhizosphere. In this study, we developed the imaging EcoFAB, a device that supports a model plant B. distachyon for 3+ weeks and is suitable for whole-root, high resolution and magnification imaging using conventional microscope system with the high degree of environmental control. Using this device, we imaged the fluorescently tagged bacterial and fungal interactions in the rhizosphere. The device accommodated solid substrates such as sand and agar in the root chamber, making it possible to study the microbes that don’t grow well in hydroponic system. Quantitative analysis of the multi-spectral image of the fluorescently tagged P. simiae strains demonstrated imaging EcoFAB’s potential to drive more reproducible and quantitative research of the field-relevant synthetic rhizosphere community. With the continual improvement in imaging system, imaging analysis algorithm, and fluorescent probes, imaging EcoFAB can be a powerful platform to interrogate and dissect the complex plant-microbial community dynamics in rhizosphere.”